# Effects of Nb Additions and Heat Treatments on the Microstructure, Hardness and Wear Resistance of CuNiCrSiCoTiNb_x_ High-Entropy Alloys

**DOI:** 10.3390/e24091195

**Published:** 2022-08-26

**Authors:** Denis Ariel Avila-Salgado, Arturo Juárez-Hernández, María Lara Banda, Arnoldo Bedolla-Jacuinde, Francisco V. Guerra

**Affiliations:** 1Facultad de Ingeniería Mecánica y Eléctrica (FIME), Universidad Autónoma de Nuevo León, Av. Universidad S/N, San Nicolás de los Garza 66450, Mexico; 2Centro de Investigación e Innovación en Ingeniería Aeronáutica (CIIIA), Facultad de Ingeniería Mecánica y Eléctrica (FIME), Universidad Autónoma de Nuevo León, Carretera a Salinas Victoria Km. 2.3, Apodaca 66600, Mexico; 3Instituto de Investigación en Metalurgia y de Materiales, Universidad Michoacana de San Nicolás de Hidalgo J. Múgica S/N, Morelia 58030, Mexico

**Keywords:** high entropy alloys, Nb additions, heat treatments, microstructure, solid solution, precipitates, hardness, wear resistance

## Abstract

In this research, a set of CuNiCrSiCoTi (H-0Nb), CuNiCrSiCoTiNb_0.5_ (H-0.5Nb) and CuNiCrSiCoTiNb_1_ (H-1Nb) high-entropy alloys (HEAs) were melted in a vacuum induction furnace. The effects of Nb additions on the microstructure, hardness, and wear behavior of these HEAs (compared with a CuBe commercial alloy) in the as-cast (AC) condition, and after solution (SHT) and aging (AT) heat treatments, were investigated using X-ray diffraction, optical microscopy, and electron microscopy. A ball-on-disc configuration tribometer was used to study wear behavior. XRD and SEM results showed that an increase in Nb additions and modification by heat treatment (HT) favored the formation of BCC and FCC crystal structures (CS), dendritic regions, and the precipitation of phases that promoted microstructure refinement during solidification. Increases in hardness of HEA systems were recorded after heat treatment and Nb additions. Maximum hardness values were recorded for the H-1Nb alloy with measured increases from 107.53 HRB (AC) to 112.98 HRB, and from 1104 HV to 1230 HV (aged for 60 min). However, the increase in hardness caused by Nb additions did not contribute to wear resistance response. This can be attributed to a high distribution of precipitated phases rich in high-hardness NiSiTi and CrSi. Finally, the H-0Nb alloy exhibited the best wear resistance behavior in the aged condition of 30 min, with a material loss of 0.92 mm^3^.

## 1. Introduction

The traditional way to design and manufacture most conventional alloys involves choosing a base element and adding other elements in specific amounts to obtain the desired properties and characteristics of the final product.

Researchers developed this approach, using Cu as a base element with additions of Ni, Cr, Si and Co. Today, there is a new technique for designing alloys with unique properties known as high-entropy alloys (HEAs).

In a broad sense, Yeh et al. [1] and Cantor et al. [2] established this new concept in 2004, by defining HEAs as alloys composed of multiple alloying elements, typically five or more, with similar atomic proportions, in a range of 5 to 35 at%, to obtain the advantages of the high entropy of the mixture, whose value should be at least 1.5R [3,4].

Table 1 shows the maximum values of configurational entropy (Δ*S*_Conf_) for equimolar multicomponent alloys with up to 11 alloying elements.

HEAs are a new class of alloys with different main elements of equiatomic or quasi-equiatomic composition. They differ from conventional alloys mainly due to the presence of a selection of complex phases due to the high configurational entropy and distortion of the crystal network [5,6].

There are thousands of combinations of elements which may be used to create HEAs, specifically metallic alloys based on the periodic table of elements [7,8]; however, those alloys with potential engineering applications should be studies especially carefully, by analyzing the properties of each element in the alloy. This analysis should consider the crystal structure, atomic size ratio, electronegativity, and the concentration of valence electrons as critical parameters, to obtain HEAs with optimal properties.

As a result, an Δ*S*_Conf_ contributes to reducing the formation of potentially brittle intermetallic compounds and stabilizes random phases in solid solution with simple structures. Hence, a reduction in the number of phases is favored, producing, as a result, HEAs with crystal structures from the systems BCC, FCC, and HCP [9,10,11].

In recent years, HEAs have been the subject of much research and development [12], attracting attention from the scientific community, because these alloys offer the potential for both high performance and a good balance of properties [13,14,15]. This kind of alloy has been reported to exhibits high hardness and wear resistance [16,17,18], high corrosion and high-temperature resistance [19], high ductility [20], good fatigue resistance and fracture toughness [21], compared with some conventional alloys [22]. 

Nevertheless, the manufacturing cost of HEAs is typically higher than for traditional alloys due to their requirement of expensive elements (e.g., Co, Be), This said, HEAs still offer a better price-to-performance ratio compared with most superalloys with high concentrations of Ni and Ti. Therefore, HEAs offer high potential value for different applications such as molds, plungers, tools and structural components exposed to wear at high temperatures.

In this context, to obtain the desired properties from HEAs, it is necessary to identify appropriate constituent elements, adequate processing routes and microstructural reinforcing methods such as solid solution, precipitation by aging, and particle dispersion. 

Most HEAs contain transition metals, particularly Ni, Co, Fe, Mn, Cr, V, and Ti [23]. These elements represent a wide range of compatibilities that allow for the formation of solid solutions and precipitated phases, to obtain alloys with attractive properties.

To this end, the present study proposes the design of new Cu-based alloys containing Ni, Co, Cr, Ni, Si, Ti, and Nb to obtain HEAs reinforced by heat treatment (HT).

As the main constituent element, Cu is responsible for high thermal conductivity, good machinability, and corrosion resistance; and exhibits a high affinity with most of the alloying elements [24]. Some of these elements promote the formation of new phases, which greatly improve mechanical properties, particularly hardness and wear resistance. This effect is attributed to a dislocation-blocking mechanism and the strengthening of grain boundaries.

For example, Ni and Co provide the appropriate conditions to generate a saturated solid solution and accelerate the recrystallization and precipitation process during aging, modifying the microstructure and increasing hardness [25,26,27]. 

The presence of a solid solution allows the formation of precipitated CoNi and δ-(Ni, Co)_2_Si during aging [28]. In the same way, Cr promotes the refinement and reinforcing of the microstructure through precipitation and the growing of particles such as CrSi_2_, Cr_2_Si, and Cr_3_Si [29]. Furthermore, Nb [30,31] and Ti [32] contribute to grain refinement and give rise to new phases in combination with Ni and Si in the Cu matrix.

Based on the above findings, this study focuses on the synthesis of new HEAs based on CuNiCoCrSi alloys described in previous studies [33,34], with the addition of Ti and various quantities of Nb. Hence, the main purpose of this study is to analyze the effect of Nb additions and heat treatment in the CuNiCrSiCoTiNb_x_ high-entropy system. In addition, this study reveals the microstructural changes and variations in hardness and wear resistance caused by Nb and Ti. Finally, we compare the wear performance of experimental alloys with that of a commercial CuBe alloy used for similar applications and thereby demonstrate the suitability of these alloys for sliding wear applications.

## 2. Materials and Methods

A set composed of CuNiCrSiCoTi (H-0Nb), CuNiCrSiCoTiNb_0.5_ (H-0.5Nb), and CuNiCrSiCoTiNb_1_ (H-1Nb) HEAs was melted in a vacuum induction furnace with 5 kg capacity using high-purity raw materials (Cu, Ni, Co, Cr and Si with 99.99% purity). Ti (73Ti-4.7Al-22.3Fe) and Nb (65Nb-32Fe-3Si) were added as ferroalloys.

An inert argon atmosphere was created in the furnace. At 1350 °C, the molten alloys were poured into steel molds with a rectangular shape to obtain ingots of 4.5 kg. Table 2 shows the chemical composition of the experimental alloys. Chemical analysis was carried out using X-ray fluorescence spectroscopy. Ingots were sectioned to obtain 20 × 20 × 10 mm samples using a Labotom-5 machine provided with coolant. Samples were subject to solution heat treatment (SHT) at 900 °C for 1 h, followed by water cooling at room temperature. Samples were then subjected to aging (AT) at 550 °C for 30 or 60 min, followed by air cooling at room temperature. Samples in AC, SHT and AT conditions were prepared using SiC abrasive paper and polished in the traditional way so that their microstructure, hardness and wear behavior could be analyzed.

XRD patterns were obtained using a PANalytical EMPYREAN diffractometer with monochromatic Co Kα_1_ radiation (wavelength = 1.7890 Å), with voltage and current of 40 kV and 40 mA, respectively, in a 2θ range from 20° to 120°, before and after heat treatment. 

Microstructure was analyzed by OM as well as by SEM using a Jeol JSM 6510LV (Jeol Ltd., Akishima, Tokyo, Japan) microscope operated at 20 kV provided with EDS detector for microanalysis. Hardness measurements were taken using a Rockwell TH 320-INCOR (Mitutoyo Corporation Ltd., Kawasaki, Kanagawa, Japan) hardness tester applying a load of 980 N for 10 S; microhardness was measured using a Vickers 402MVD tester applying a load of 0.025 N for 15 s, creating a total of 15 indentations for sample and experimental conditions. Dry sliding wear performance and friction coefficients were evaluated using a ball-on-disc test (Ball on Disc Wear Testing Machine, UANL, Monterrey, Nuevo León, México) applying a normal load of 30 N at room temperature. A G25 steel ball of 11.1 mm diameter was used as a counterface according to the ASTM-G99-95 standard [35]. The test parameters were as follows: an initial radius of 6 mm for the wear track; a sliding speed of 365 rpm; and sliding distances of 125 m, 250 m, 375 m, and 500 m. 

Wear tracks were digitalized with a non-contact profiler NANOVEA PS50 3D (Irvine, CA, USA). Wear losses were measured in mm^3^ using profiler measurement software. Two repetitions for each test were carried out to ensure data reliability. Worn surfaces and wear debris were analyzed by SEM-EDS.

## 3. Thermodynamic Parameters and Considerations for Phase Formation in HEAs

Phase transformations and reactions to form composites are thermodynamically governed by variations in Gibbs free energy, which takes into account changes in entropy and enthalpy as functions of temperature. In contrast to the intermetallic compounds, solid solutions exhibit a higher configurational entropy (Δ*S*_Conf_) in a range of 12 ≤ Δ*S*_Conf_ ≤ 17.5 J/mol·K, making them more stable at high temperatures [36]. 

Similarly, Guo and Liu reported that the formation of a solid solution requires an adequate range of Δ*S*_Conf_, Δ*H*_Mix_ and δ, corresponding to 11 ≤ Δ*S*_Conf_ ≤ 19.5 J/mol·K, −22 ≤ Δ*H*_Mix_ ≤ 7 KJ/mol, and δ ≤ 8.5%, respectively [37].

In the present study, Δ*S*_Conf_ and Δ*H*_Mix_ values fall within a range favorable to promote the formation of solid solutions; obviously, there is a higher probability of obtaining solid solutions for alloys H-0.5Nb and H-1Nb, due to their higher entropy value caused by Nb additions.

Table 3 shows the theoretical values of enthalpy for the binary mixture $\Delta H_{\left[AB\right]}^{mix}$, corresponding to the alloying elements used in the experimental alloys, to produce the resultant Δ*H*_Mix_ [38].

In addition, we require Ω parameter correlates Δ*H*_Mix_ and Δ*S*_Conf_, coupled with the atomic size difference parameter δ, to predict the stability of the solid solutions in the HEAs; to this end, we require values of Ω ≥ 1.1 and δ ≤ 6.6 [39]. By such means, the alloys of the present study are susceptible to producing solid solutions.

However, Guo at al. reported the importance of the valence electron concentration (VEC) in promoting the stability of the BCC and FCC phases. BCC phases are stable for VEC values < 6.87; conversely, FCC phases are stable for VEC values ≥ 8 [40]. Similarly, the electronegativity difference (**Δχ**) should be between 0.11 ≤ Δχ ≤ 0.17 [41]. Based on the calculations of VEC and Δχ, it is evident that H-0Nb, H-0.5Nb and H-1Nb will be predominantly FCC, and BCC only to a lesser extent.

As a consequence, in addition to the enthalpy of the mixture (Δ*H*_Mix_) and the configurational entropy (Δ*S*_Conf_), we must also consider other critical aspects that play a role in the formation of HEAs which can be predicted through thermodynamic calculations, to aid the design and production of HEAs with the desired characteristics [42]. 

Therefore, to successfully design CuNiCoCrSiTiNb_x_ alloys of high entropy for this study, it was necessary to perform an extensive analysis and calculation of the different relevant criteria that influence the feasibility of obtaining the desired phases. Configurational entropy (Δ*S*_Conf_), enthalpy of mixture (Δ*H*_Mix_), theoretical melting point (Tm), omega (Ω), atomic size difference (δ), electronegativity difference (Δχ) and valence electron concentration (VEC) together represent the main thermodynamic parameters that help to predict the formation of solid solutions, cubic crystal structures and intermetallic compounds. The equations used for the calculation of the different parameters can now be expressed as follows:(1)$$\Delta S_{\mathrm{Conf}}\;=-R\sum_{i=1}^{n}C_{i}\ln C_{i}$$
(2)$$\Delta H_{\mathrm{Mix}}=\sum_{i=1,\;j\neq i}^{n}4\Delta H_{\mathrm{ij}}C_{i}C_{j}$$
(3)$$\Omega =\frac{T_{m\;}\Delta S_{conf}\;}{\left|\Delta H_{Mix}\right|},\;T_{m}=\sum_{i=1}^{n}C_{i}{\left(T_{m}\right)}_{i}$$
(4)δ=∑i=1nCi1−rir¯2,r¯=∑i=1nCiri
(5)$$\Delta \chi =\sqrt{\sum_{i=1}^{n}C_{i}{\left(\chi_{i}-\bar{\chi }\right)}^{2}},\bar{\chi }=\sum_{i=1}^{n}C_{i}\chi_{i}$$
(6)$$\mathrm{VEC}=\sum_{i=1}^{n}C_{i}{\left(VEC\right)}_{i}$$
where *R* is the gas constant (8.3145 J/K·mol); *C_i_* and *C_j_* are the atomic concentrations of the *i*th and *j*th atom; Δ*H_ij_* is the binary mixing enthalpy AB for multicomponent alloys; *T_m_* is the theoretical melting temperature of the mixture; *r_i_* is the atomic ratio of each element; and χ*_i_* is the Pauling electronegativity for each element.

The corresponding values of the different parameters calculated with the above equations for the H-0Nb, H-0.5Nb, and H-1Nb alloys are shown in Table 4.

## 4. Results and Discussion

### 4.1. Phase Structure by XRD before Heat Treatment

Figure 1 shows the XRD patterns of the HEAs in the as-cast condition. Figure 1a shows the 2θ range from 35° to 125°, whereas Figure 1b shows a magnification from 48° to 55° for the (103), (111), (210), and (121) peaks. From Figure 1, we can clearly see that the alloys H-0.5Nb and H-1Nb mainly exhibit FCC and BCC phases. It can also be seen that alloy H-0Nb experienced high intensity at the (111) peak (see Figure 1b). This effect can be attributed to the preferential orientation of the matrix rich in Cu content. In addition, Figure 1b shows that the (111) peak involves a significant change in the diffraction angle, which is displaced to the right from 50.64° to 50.69°, and then to 50.75°, as the Nb content increases. This behavior is attributed to Nb additions which cause a reduction in the lattice parameter constant from 3.63 Å for the H-0Nb alloy to 3.62 Å for H-0.5Nb, and 3.61 Å in the case of H-1Nb. These lattice parameter values were calculated from the XRD results using Equations (7)–(9).
2dsinθ = λ,(7)
(8)$$d=a/\sqrt{h^{2}+{\;k}^{2}+{\;l}^{2}},$$
(9)$$a=\lambda \sqrt{h^{2}+{\;k}^{2}+{\;l}^{2}}/(2\sin\theta ),$$
where d corresponds to the interplanar spacing of the crystal; the diffraction angle θ; the wavelength λ for Co Kα_1_ (λ = 1.7890 Å); the lattice constant a; and the miller index h, k, l.

In addition, some precipitated phases were found, such as N_2_Si and Cr_3_Si, which are responsible for the microstructure strengthening reported in previous studies, where Cr/Si precipitates exhibited higher stability compared with Ni/Si precipitates at elevated temperatures [43].

### 4.2. Microstructural Characterization before Heat Treatments 

The microstructure was observed using an optical microscope to analyze Nb effects in the as-cast condition. For this purpose, samples were prepared in the traditional way, followed by etching with a solution composed of 3.3 g of FeCl_3_ in 100 mL of ethanol with 17 mL of HCl and 1 mL of HNO_3_. Samples were submerged from 1 to 5 s and quickly washed and dried [44]. Figure 2 shows representative images of the microstructures of alloys (a) H-0Nb, (b) H-0.5Nb, and (c) H-1Nb in the as-cast condition. It can be seen that the alloys show a dendritic structure composed of three phases labeled A, B, and C, respectively. Based on previous studies, we determined that the A phase corresponded to the matrix αCu, which experienced a reduction in its volume content with increased additions of Nb. This reduction causes an increase in the B and C phases rich in NiSiTi and CrSi, respectively. It can also be seen that the C phase is located at the edges of the dendritic arms of the B phase. By comparing the images in Figure 2a–c, we can also identify a refining effect on the microstructure as a result of the increase in Nb additions. It is worth noting that Nb does fulfill its function as a grain refiner; consequently, the H-0.5Nb and H-1Nb alloys show a different microstructure to the H-0Nb alloy.

### 4.3. Phase Structure by XRD after Heat Treatments

Figure 3 compares XRD patterns to show the effects of heat treatment upon the alloys (a) H-0-Nb, (b) H0.5-Nb, and (c) H1-Nb. A magnification of the main peaks of Figure 3c is shown in Figure 3d. It should be noted that all the alloys present FCC and BCC phases. 

Figure 3a shows an obvious decrease in the intensity of the main (111) peak due to SHT and AT heat treatments. This can be attributed to a lower crystallographic orientation. In addition, samples subject to SHT experienced a lower intensity of most diffraction peaks, suggesting a better distribution of phases and elements in solution with the αCu matrix due to the temperature and holding time in solubilizing heat treatment.

The similarity of peaks subjected to AT-60, shown in Figure 3a,b, should also be noted. The diffraction patterns shown in Figure 3c,d enable a better analysis of the effects of SHT and AT upon the alloy H-1Nb, which reveals a significant difference between H-0Nb and H-0.5Nb alloys. Another significant difference between the AT-30 and AT-60 samples is the decreased intensity of most diffraction peaks, particularly the (111) peak, which also exhibits a slight displacement to a lower angle [see Figure 3d], from 50.69° to 50.60°, with a corresponding increase in the lattice parameter from 3.62 Å to 3.63 Å. This behavior can be attributed to the increase in interplanar spacing caused by thermal expansion at the aging temperature. In this sense, for the H-1Nb alloy, the addition of 1 Wt% of Nb and HT modification results in conditions favorable for obtaining new phases. As a result, the presence of such precipitated phases as Co_2_Nb, N_2_Si, Cr_3_Si and Nb_6_Ni_16_Si_7_ was confirmed.

### 4.4. Microstructural Characterization after Heat Treatments

Figure 4 shows a sequence of SEM in a secondary electron mode (SEI) for the H-1Nb alloy under different experimental conditions as follows: (a) as-cast, after heat treatment; (b) SHT; (c) AT-30; and (d) AT-60.

The microstructures show three different tones, as follows: light gray regions, labeled as A; dark gray regions, labeled as B; and black regions, labeled as C.

When comparing Figure 4a,b, a higher amount of Cu-rich A and NiSiTi-rich B phases and a lower amount of CrSi C phase can be identified. In addition, Figure 4b shows a higher amount of B phase due to the high temperature, holding time, and fast cooling rate after SHT.

Figure 4c,d also show the evolution of the microstructure, which experiences an increase in precipitation with an increase in aging holding time from 30 to 60 min. Element distribution in the different phases is shown in Figure 5, Figure 6 and Figure 7 in the next section.

### 4.5. Microstructural Characterization by SEM-EDS 

Figure 5a shows a backscattered electron image of the H-0Nb alloy in which (b–h) correspond to elemental mapping for (b) Cu Kα; (c) Si Kα; (d) Cr Kα; (e) Ni Kα; (f) Ti Kα; (g) Co Kα; and (h) Fe Kα from the same area. Similarly, Figure 6 and Figure 7 show the EDS elemental mapping for alloys H-0.5Nb and H-1Nb, with Nb distribution also shown in Figure 6i and Figure 7i. As can be seen, there is a change in phase distribution, as previously shown in Figure 2 and Figure 4; however, element distribution based on the EDS mapping does not reveal any significant difference between alloys. Figure 5, Figure 6 and Figure 7a show that the A phase is mainly composed of Cu, while phase C is mainly composed of Cr. Additionally, Table 5 shows the average results of 10 EDS punctual microanalyses performed in each phase in the AC condition. As can be seen, in the A phase, Cu corresponds to 82.85 at%. In the case of B, the sum of NiSiTi represents 75.67 at%. For the C phase, the total for CrSi is 93.32 at%. Table 5 also shows significant differences in the composition of the phases of the three alloys in the AC condition.

### 4.6. Macrohardness Rockwell B and Microhardness Vickers Test Results 

Figure 8a shows a graph with hardness values for the CuBe commercial alloy and the H-0Nb, H-0.5Nb, and H-1Nb experimental alloys in the AC, SHT, and AT conditions on the HRB scale. Similarly, Figure 8b shows the average values of (HV) for the different phases in alloys H-0Nb, H-0.5Nb, and H-1Nb.

Figure 8a shows increased hardness values for AC alloys with increases in Nb addition from 103.4 ± 0.24 HRB to 106.58 ± 0.12 HRB for the alloy H-0Nb, and to 107.53 ± 0.11 HRB for both H-0.5Nb and H-1Nb. This increase in hardness is attributed to a higher volume content of B and C phases and a consequent decrease in A as shown in the microstructural characterization. We should also note that this increase in hardness with respect to Nb addition is consistent for all experimental conditions. A general decrease in hardness is observed for SHT samples compared with the AC condition, with obtained values of 98.14 ± 0.25, 100.87 ± 0.11, and 102.22 ± 0.39 HRB for the H-0Nb, H-0.5Nb, and H-1Nb alloys, respectively. this behavior is attributed to a slight reduction in content of B and C phases due to the increase in solubility of the different alloying elements in the αCu matrix (A). Samples subject to AT exhibited an increase in hardness compared with AC and SHT conditions, with obtained values of 110.5 ± 0.15 HRB, 112.43 ± 0.18 HRB and 112.98 ± 0.08 HRB for the H-0Nb, H-0.5Nb and H-1Nb alloys, respectively, in the AT-60 condition. This hardness increase is attributed to precipitation within the matrix of Cr_3_Si and Ni_2_Si. The above hardness values are considerably higher when compared with an age-hardened CuBe commercial alloy with a hardness value of around 83.47 ± 2 HRB.

The Vickers microhardness values presented in Figure 8b confirm previous observations where the Cu-rich A phase produces lower microhardness values, while the NiSiTi-rich B phase, and the CrSi rich C phase produce higher hardness values of approximately 930 and 1200 HV, respectively. We find, therefore, that hardness changes are mainly determined by the variation in the volume content of phases, by their distribution (refinement), and by the precipitation of particles that produces a strengthening of the αCu matrix during AT.

### 4.7. Wear and Friction Behaviors of the HEAs

Wear losses as a function of the sliding distance for intervals of 125 m values are plotted in Figure 9 which shows a consistent increase in wear losses as a function of the sliding distance. However, wear losses increased with Nb addition. In the as-cast condition, H-0Nb showed a volume loss of 0.92 mm^3^ and arithmetic mean roughness (Ra) of 0.059 µm, while the Nb-added alloys exhibited volume losses of 1.19 mm^3^ and 1.33 mm^3^, and Ra values of 0.69 µm and 0.73 µm, for the H-0.5Nb and H-1Nb alloys, respectively. This behavior might seem contrary to traditional wear theories where higher levels of hardness accompany a reduction in wear losses; however, in this case, the presence of a higher volume fraction of hard-but-brittle phases causes an increase in the roughness values and friction coefficients, as can be seen in Figure 10, where the increase in the friction coefficient is associated with a lower volume content of the αCu matrix. It is well known that the couple formed by Cu/steel exhibits a low friction coefficient, as can be seen in Figure 10, where the CuBe alloy exhibits the lowest friction coefficient. However, this alloy also exhibited the highest volume losses due to a higher degree of deformation and delamination caused by surface fatigue, resulting in a value of 3.34 mm^3^. These findings highlight the importance of a good balance between hardness, distribution, and tribological properties. Figure 9 and Figure 10 also show that the AT-30 condition produces lower values of wear losses and friction coefficients for all alloys. In this condition, the H-0Nb alloy exhibited a volume loss of 0.92 mm^3^, while values of 0.96 mm^3^ and 0.99 mm^3^, were recorded for the H-0.5Nb and H-1Nb alloys, respectively, and friction coefficient levels were consistent with these values. For the AT-60 condition, a slight increase in the friction coefficient can be observed, accompanied by an increase in wear losses. In this case, the higher number of precipitated phases within the matrix caused a higher discontinuity of the oxide layer, facilitating its fracture and detachment and increasing the metal-to-metal contact.

Figure 11 shows a sequence of SEM images obtained from the worn surfaces of the H-0Nb [Figure 11a], H-0.5Nb [Figure 11b], and H-1Nb [Figure 11c] alloys, and also the CuBe [Figure 11d] commercial alloy, after they were subjected to the wear test in dry conditions. Here, we see greater damage on the surface with the increase in Nb addition, with the H-0Nb alloy exhibiting lower surface damage than the H-0.5Nb and H-1Nb alloys. Similarly, the CuBe commercial alloy also showed considerable surface damage, as can be seen on the 3D digitalized images of worn surfaces and profiles shown in Figure 12a–d. The higher degree of deformation and wear damage observed in the CuBe and Nb-added alloys can be attributed to the detachment of hard abrasive particles changing the wear mechanism from sliding wear to abrasive wear.

In addition, the EDS elemental mapping images of Figure 13 show a lower discontinuity of the oxide layer in the alloy H-0Nb [Figure 13a] compared with H-0.5Nb [Figure 13b]. As mentioned before, the higher number of high-hardness precipitates embedded in the soft and ductile matrix causes a higher degree of abrasion and a higher discontinuity of the oxide layer, resulting in fracture and detachment, increasing the metal-to-metal contact. These observations agree with the friction coefficient values previously shown in Figure 10, and with the observations reported by different authors [45,46], because the presence and stability of this oxide layer are responsible for reducing the friction coefficient by inhibiting the metal-to-metal contact and thus reducing wear losses.

Figure 14 shows the SEM micrographs of the wear particles obtained after finishing the wear tests at 500 m distance. The SEM micrographs evidence a higher particle detachment of greater size as Nb increases; however, the commercial CuBe alloy exhibits higher particle detachment of greater size due to wear. In addition, EDS punctual microanalysis in Figure 14 shows evidence of oxides and lower degrees of oxygen concentration in the alloys with increases in Nb.

Finally, Figure 15 shows a 3D digitalized image of the steel balls used for the wear test. In Figure 15a, which corresponds to the ball used to test the H-0Nb alloy, the 3D view shows evidence of slight wear damage; in contrast, in Figure 15b, which corresponds to the ball used for the CuBe alloy test, a rougher surface and certain degree of adhesion can be observed. 

These experimental results show evidence of the fracture and detachment of the oxide layer responsible for protecting the bare surface from contact with metals, resulting in high contact temperatures which promote the adhesive wear. In addition, once this layer is broken and detached, abrasive particles between the bare surfaces increase the friction coefficient, producing greater wear damage, as observed in the case of the CuBe and Nb-added alloys.

## 5. Conclusions

Analysis of the Nb additions and heat treatment effects on microstructure, phase transformation, hardness, and wear resistance compared with a commercial CuBe showed the following:(1)Nb addition resulted in a gradual increase in configurational entropy from 12.14 J/mol·K 12.36 J/mol·K in the H-0Nb alloy, and to 12.55 J/mol·K in both the H-0.5Nb and H-1Nb alloys, resulting in the formation of FCC and BCC solid solutions.(2)DRX results revealed the presence of FCC and BCC phases as well as Co_2_Nb, N_2_Si, Cr_3_Si and Nb_6_Ni_16_Si_7_ compounds. In addition, calculations of the lattice constant showed a reduction based on the displacement of the (111) peak to higher angles as Nb levels increased in the alloys.(3)Microstructure transformation was influenced by Nb additions and heat treatment producing the precipitation of interdendritic phases and microstructure refinement.(4)The increase in hardness and microhardness of the HEAs were attributed to a higher content of NiSiTi and CrSi-rich phases of high hardness as well as to the precipitation of particles during AT, resulting in higher hardness values, compared with the CuBe commercial alloy.(5)The H-0Nb alloy exhibited better wear performance for all experimental conditions. However, wear losses also increased with Nb addition. This was attributed to microstructure fragilization due to a high-density Cr_3_Si precipitation of high hardness. The best wear performance was exhibited by the H-0Nb alloy in the AT-30 condition, with a volume loss of 0.92 mm^3^, which very low when compared to the value of 3.34 mm^3^ obtained for the CuBe commercial alloy.

## Figures and Tables

**Figure 1 entropy-24-01195-f001:**
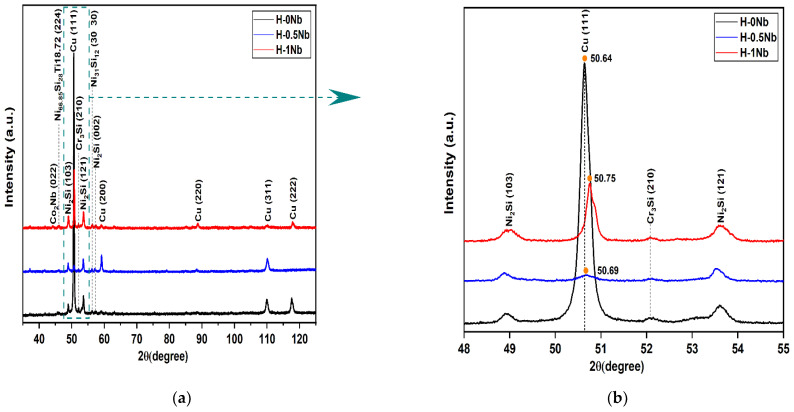
(**a**) XRD patterns of the HEAs: H-0Nb, H-0.5Nb, and H-1Nb alloy in as-cast condition; (**b**) some enlarged peaks of (**a**).

**Figure 2 entropy-24-01195-f002:**
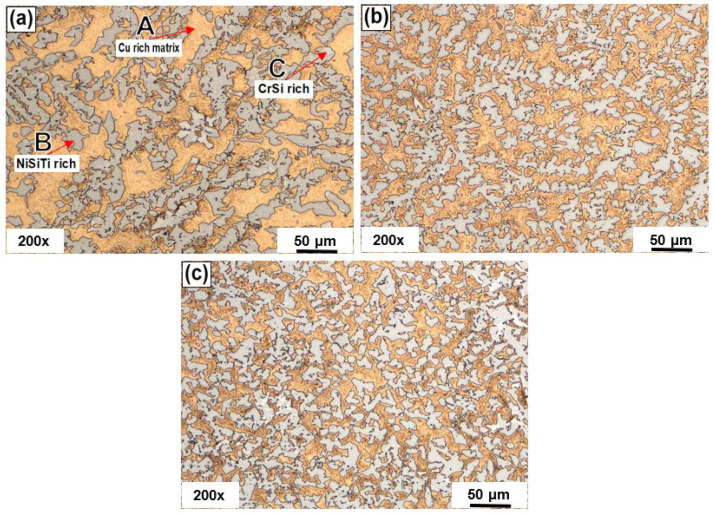
Optical microstructure of HEAs in as-cast condition: (**a**) H-0Nb, (**b**) H-0.5Nb, and (**c**) H-1Nb.

**Figure 3 entropy-24-01195-f003:**
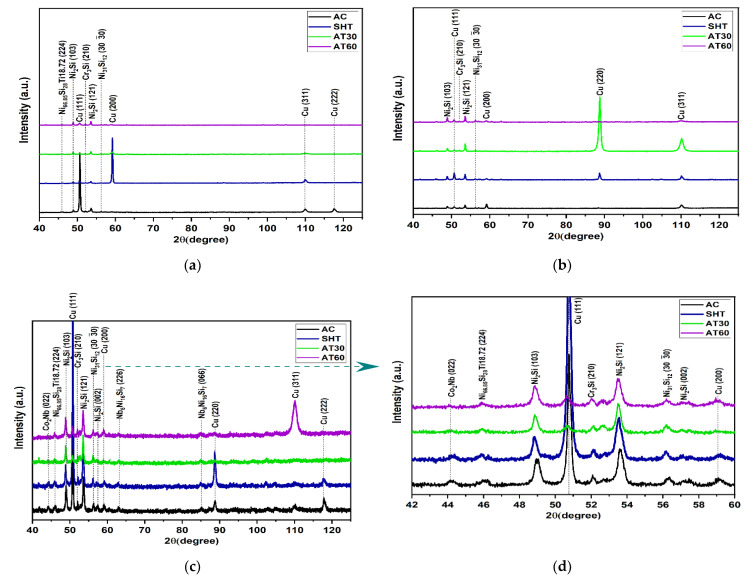
XRD patterns of HEAs: (**a**) H-0Nb, (**b**) H-0.5Nb, and (**c**) H-1Nb alloy, before and after heat treatments; and (**d**) the enlarged peaks of (**c**).

**Figure 4 entropy-24-01195-f004:**
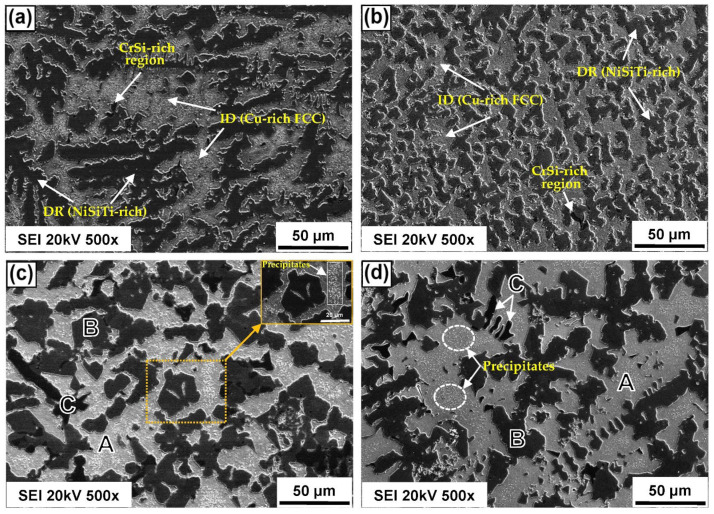
SEM micrographs for H-1Nb alloy in conditions: (**a**) as-cast; (**b**) SHT; (**c**) AT-30; and (**d**) AT-60, respectively.

**Figure 5 entropy-24-01195-f005:**
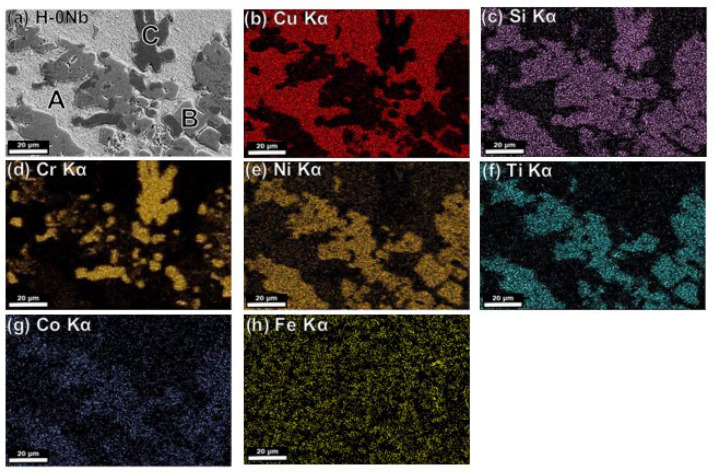
(**a**) SEM-BSE image of H-0Nb alloy in as-cast condition, and SEM-EDS elemental mapping images for (**b**) Cu Kα; (**c**) Si Kα; (**d**) Cr Kα; (**e**) Ni Kα; (**f**) Ti Kα; (**g**) Co Kα; (**h**) Fe Kα from the same area as (**a**).

**Figure 6 entropy-24-01195-f006:**
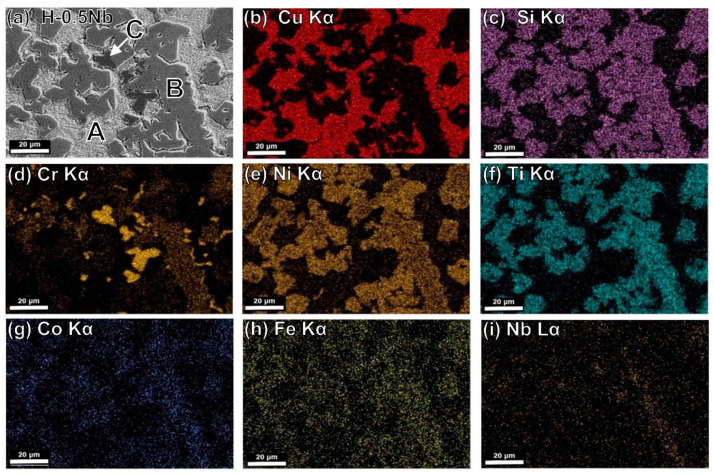
(**a**) SEM-BSE image of H-0.5Nb alloy in as-cast condition, and SEM-EDS elemental mapping images for (**b**) Cu Kα; (**c**) Si Kα; (**d**) Cr Kα; (**e**) Ni Kα; (**f**) Ti Kα; (**g**) Co Kα; (**h**) Fe Kα; (**i**) Nb Lα from the same area as (**a**).

**Figure 7 entropy-24-01195-f007:**
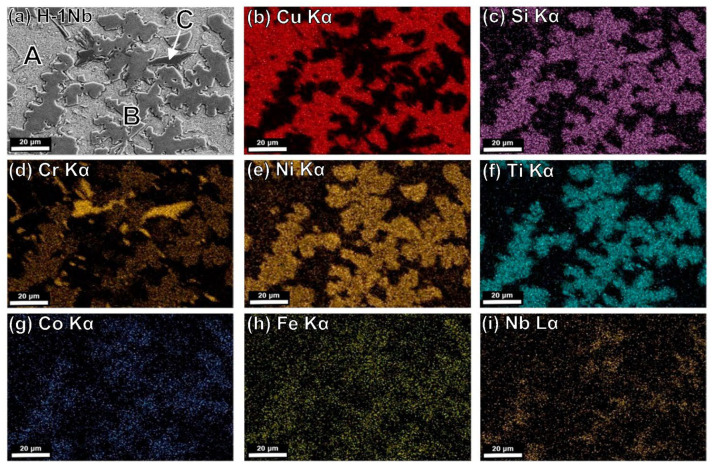
(**a**) SEM-BSE image of H-1Nb alloy in as-cast condition, and SEM-EDS elemental mapping images for (**b**) Cu Kα; (**c**) Si Kα; (**d**) Cr Kα; (**e**) Ni Kα; (**f**) Ti Kα; (**g**) Co Kα; (**h**) Fe Kα; (**i**) Nb Lα from the same area as (**a**).

**Figure 8 entropy-24-01195-f008:**
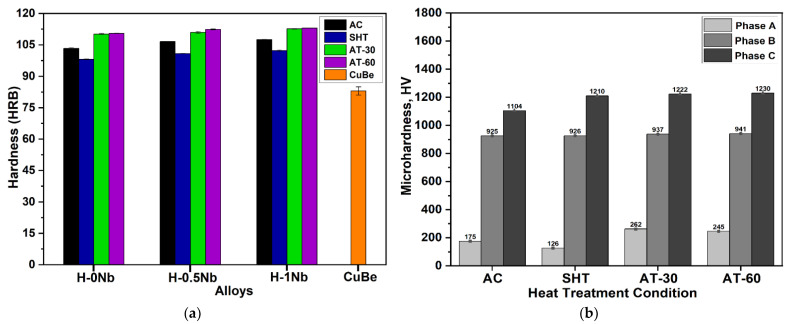
Hardness results for H-0Nb, H-0.5Nb, and H-1Nb alloys before (AC) and after heat treatments (SHT, AT-30, AT-60): (**a**) Macrohardness Rockwell B (HRB); and (**b**) average microhardness Vickers (HV) test result values of individual phases.

**Figure 9 entropy-24-01195-f009:**
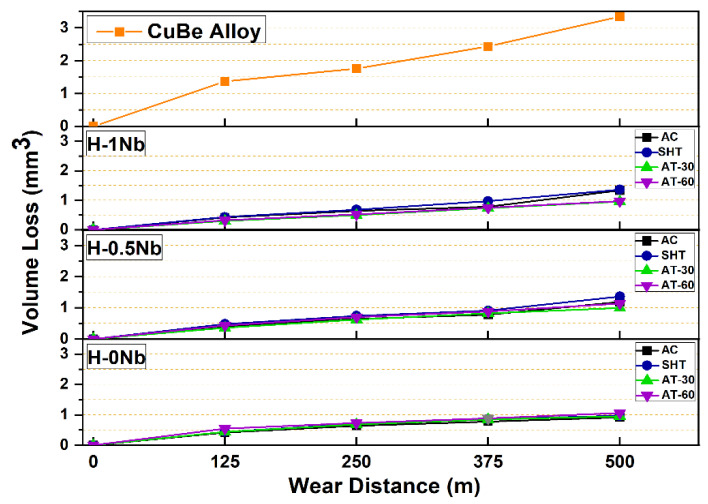
Wear test results for H-0Nb, H-0.5Nb, and H-1Nb alloys before (AC) and after heat treatments (SHT, AT-30, AT-60); and for CuBe commercial alloy.

**Figure 10 entropy-24-01195-f010:**
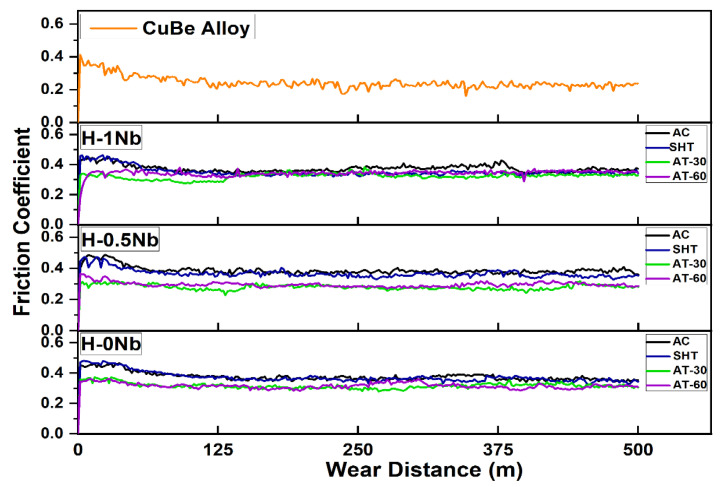
Variation of friction coefficient obtained at the surface of H-0Nb, H-0.5Nb, and H-1Nb alloys before (AC) and after heat treatments (SHT, AT-30, AT-60) and for CuBe commercial alloy.

**Figure 11 entropy-24-01195-f011:**
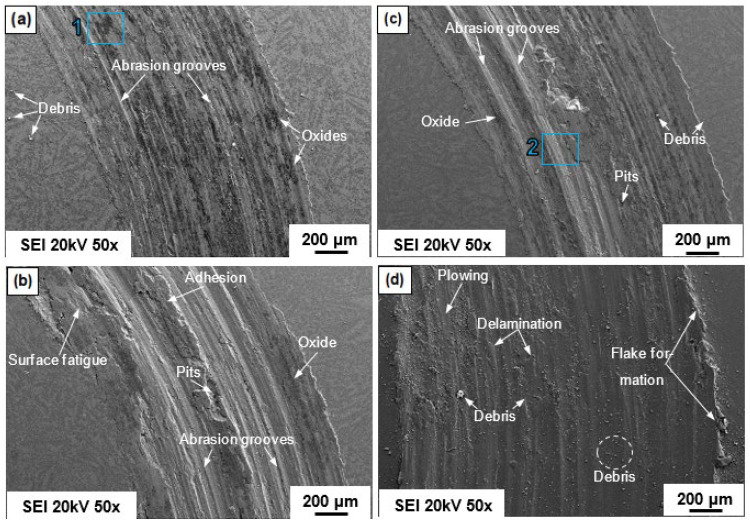
SEM images (SEI) obtained after the wear test at a sliding distance of 500 m on the worn surfaces of alloys (**a**) H-0Nb; (**b**) H-0.5Nb; and (**c**) H-1Nb, in as-cast condition; and (**d**) the CuBe commercial alloy.

**Figure 12 entropy-24-01195-f012:**
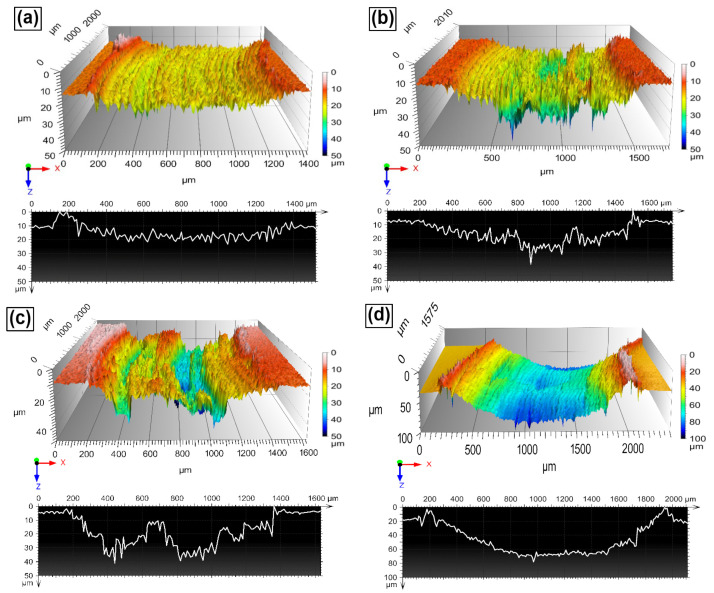
3D morphology and cross-section profiles obtained after the wear test at a sliding distance of 500 m on the worn surfaces of the (**a**) H-0Nb, (**b**) H-0.5Nb, (**c**) H-1Nb alloys in as-cast condition; and (**d**) the CuBe commercial alloy.

**Figure 13 entropy-24-01195-f013:**
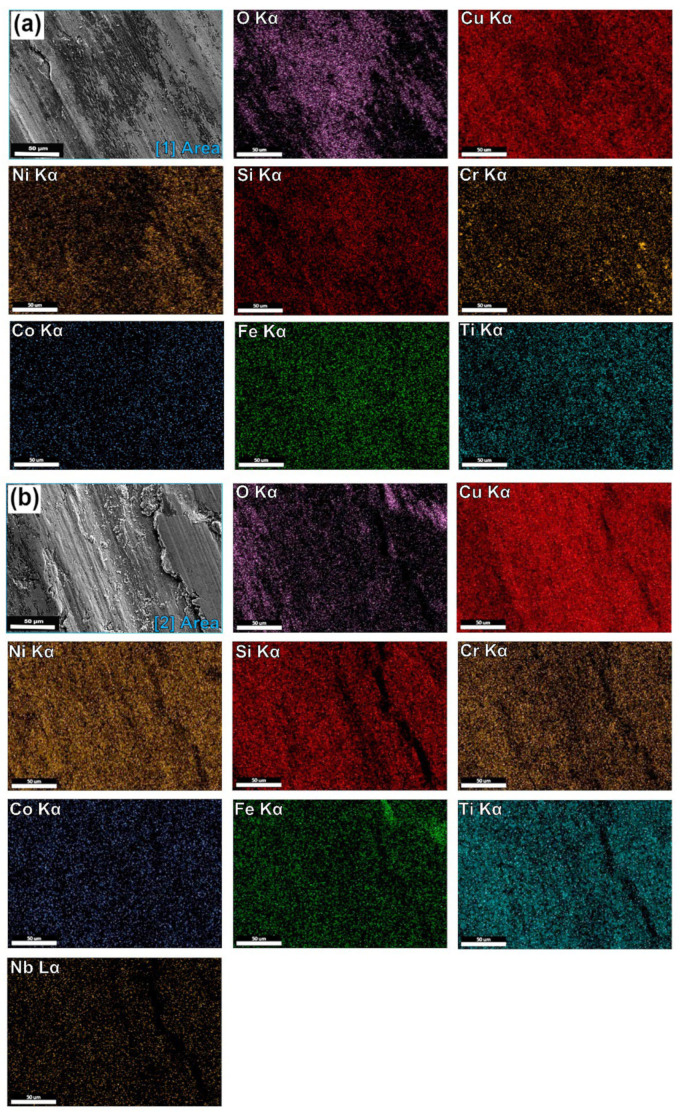
SEM images of wear marks and SEM-EDS elemental mapping images of oxide scale formed on Area 1 of (**a**) H-0Nb alloy and Area 2 of (**b**) H-0.5Nb alloy after the wear test from the same areas shown in Figure 11a,b.

**Figure 14 entropy-24-01195-f014:**
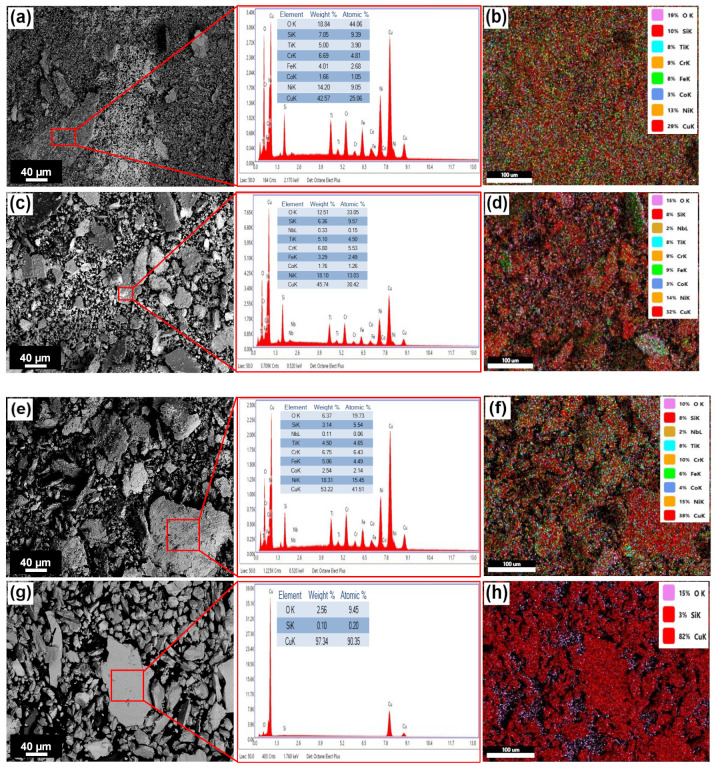
SEM micrographs and EDS mapping spectrum area of the wear debris collected from the samples in as-cast condition: (**a**,**b**) H-0Nb; (**c**,**d**) H-Nb0.5; (**e**,**f**) H-Nb1; and (**g**,**h**) CuBe alloy.

**Figure 15 entropy-24-01195-f015:**
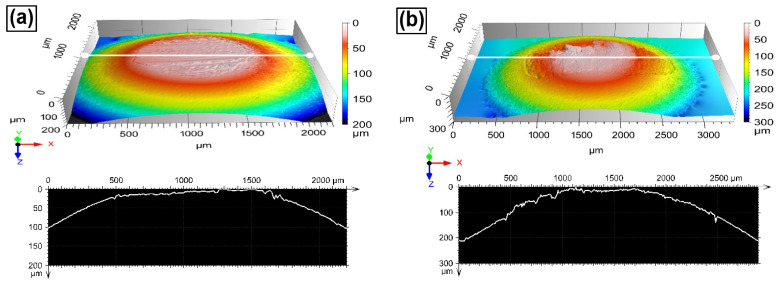
3D morphology of the balls and cross-section profiles obtained by profilometer after the wear test at a sliding distance of 500 m from the surface of the balls sliding on the samples: (**a**) H-0Nb alloy in as-cast condition; and (**b**) CuBe commercial alloy.

**Table 1 entropy-24-01195-t001:** Configurational entropies of equimolar alloys with constituent elements up to 11.

*n*	*1*	*2*	*3*	*4*	*5*	*6*	*7*	*8*	*9*	*10*	*11*
**Δ*S*_Conf_**	0	0.69*R*	1.1*R*	1.39*R*	1.61*R*	1.79*R*	1.95*R*	2.08*R*	2.2*R*	2.3*R*	2.4*R*

**Table 2 entropy-24-01195-t002:** Chemical compositions of the developed HEAs.

Alloy	Elemental Constituents (wt%)							
	Cu	Ni	Cr	Si	Co	Ti	Fe	Nb
H-0Nb	56.92	19.50	6.50	6.50	2.00	6.50	2.08	0.00
H-0.5Nb	56.18	19.50	6.50	6.50	2.00	6.50	2.32	0.50
H-1Nb	55.44	19.50	6.50	6.50	2.00	6.50	2.56	1.00

**Table 3 entropy-24-01195-t003:** Enthalpy values of $\Delta H_{\left[AB\right]}^{mix}$ (KJ/mol) for the elements of Cu, Ni, Cr, Si, Co, Ti, Fe, and Nb.

Element (Atomic Size, NM)	Cu	Ni	Cr	Si	Co	Ti	Fe	Nb
Cu (0.135)	-	4	12	−19	6	−9	13	3
Ni (0.135)	-	-	−7	−40	0	−35	−2	−30
Cr (0.140)	-	-	-	−37	−4	−7	−1	−7
Si (0.110)	-	-	-	-	−38	−66	−35	−56
Co (0.135)	-	-	-	-	-	−28	−1	−25
Ti (0.14)	-	-	-	-	-	-	−17	2
Fe (0.14)	-	-	-	-	-	-	-	−16
Nb (0.145)	-	-	-	-	-	-	-	-

**Table 4 entropy-24-01195-t004:** Thermodynamic parameters for phase formation in HEAs.

Alloy	Δ*S*_Conf_ (J/mol·K)	Δ*H*_Mix_ (k·J/mol)	Ω	Tm (K)	δ (%)	Δχ (%)	VEC	CS
H-0Nb	12.14	−13.66	1.35	1516.15	5.44	0.110	8.93	FCC/BCC
H-0.5Nb	12.36	−13.81	1.36	1519.15	5.49	0.111	8.90	FCC/BCC
H-1Nb	12.55	−13.95	1.37	1522.15	5.53	0.111	8.87	FCC/BCC

**Table 5 entropy-24-01195-t005:** The average chemical composition for each alloy obtained by SEM-EDS for individual phases in the AC condition.

Alloy	Phase	Elemental Constituents (At%)							
		Cu	Si	Cr	Ni	Ti	Co	Fe	Nb
H-0Nb	Nominal	50.00	12.93	6.98	18.54	7.58	1.89	2.08	0.00
	A	82.85	8.57	0.00	6.86	0.23	0.24	1.25	0.00
	B	10.21	24.42	5.06	39.94	12.31	3.97	4.09	0.00
	C	1.48	23.02	70.30	1.14	0.42	0.00	3.64	0.00
H-0.5Nb	Nominal	49.40	12.94	6.98	18.56	7.59	1.90	2.33	0.30
	A	85.74	6.37	0.00	6.56	0.26	0.28	0.79	0.00
	B	4.59	23.92	5.58	43.80	14.44	3.90	3.33	0.44
	C	0.87	22.21	70.53	1.63	0.65	0.63	3.45	0.03
H-1Nb	Nominal	48.81	12.96	6.99	18.59	7.60	1.90	2.55	0.60
	A	84.49	6.09	0.00	7.34	0.53	0.35	1.20	0.00
	B	5.44	23.12	16.40	30.90	14.02	4.68	4.30	1.14
	C	1.31	21.35	70.00	2.15	0.20	0.66	3.74	0.59

## Data Availability

Not applicable.

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
