# Peer review of "Effects of Nb Additions and Heat Treatments on the Microstructure, Hardness and Wear Resistance of CuNiCrSiCoTiNbx High-Entropy Alloys"

_entropy, 2022, doi:10.3390/e24091195_

Round 1

Reviewer 1 Report

The Paper is readable and well written.

Comments and suggestions:

1. Introduction

Line 81 (change y into and);  Most HEAs contain transition metals, particularly Ni, Co, Fe, Mn, Cr, V y Ti

Into

Most HEAs contain transition metals, particularly Ni, Co, Fe, Mn, Cr, V, and Ti

Line 85 (change y into and): For this purpose, the present study exposes the design of new Cu-based alloys containing Ni, Co, Cr, Ni, Si, Ti y Nb to obtain HEAs reinforced by heat-treatment (HT).

Into

For this purpose, the present study exposes the design of new Cu-based alloys containing Ni, Co, Cr, Ni, Si, Ti, and Nb to obtain HEAs reinforced by heat-treatment (HT).

Lines 95 and 97 same as for lines 81, and 85 (change y into and)

2. Materials and Methods

Please provide additional information about:

Line 114; Methods used for chemical composition determination, presented in Table 2.

Was the base composition measured from the first heat preparation and then used for the additional two heats with Nb additions? Any alloying schedule?

Line 112; When was Nb added regarding metal melt preparation? At the beginning of the melt or just before pouring? A similar goes for Ti, due to the potential fading of the element. Did the authors check if Nb was completely dissolved in the metal melt, as this can be an issue, especially at such low temperatures for given ferroalloy due to the sluggish dissolving process?

Line 119 (reconsider);

Samples in AC, SHT, and AT conditions were sanded with SiC abrasive paper…

Into

Samples in AC, SHT, and AT conditions were prepared using SiC abrasive paper…

Line 123 (change Kv into kV)

3. Thermodynamic parameters and considerations for phase formation in HEAs alloys

Line 171 (cualculation into calculation );

Line 192;

Authors should emphasize that equation for Tm (eq. 3) is indicative only. Using the mechanical mixture rule (eq. 3), this is not strictly straightforward as the solidus surface is not flat but curved. The CALPHAD approach could also be used for Tm prediction by using a proper thermodynamic database in this case.

4. Results and Discussion

  Line 258-273: Did the authors check if interdendritic precipitation inside the Cu-rich matrix was enhanced by increasing Nb in the as-cast state? Based on OM in Figure 2, it seems that the intra-dendritic precipitation could be enhanced with Nb addition in the as-cast state.

Line 312; Figure 3 (a) instead of Figure 1 (a).

Line 360;  Based on the statement “As can be seen by comparing Figures 4 (a) and (b), there is a higher content of Cu-rich A and NiSiTi-rich B phases and a lesser amount of CrSi C phase

Have authors done quantitative areal fraction analysis based on OM or SEM micrographs for phases such as CrSi, etc. or it is based on a visual assessment?

Author Response

Dear Reviewer,                                                                       August 20, 2022

I am very grateful for your comments and observations. Authors are agree with comments and observations, every comment and change has been accepted and corrected.

Reviewer 2 Report

The manuscript "Effects of Nb Additions and Heat Treatments on the Microstructure, Hardness and Wear Resistance of CuNiCrSiCoTiNbx High Entropy Alloys" is devoted to study 7-elements alloy. The presence of Nb effects on the properties of the studied material and changes of hardness and wear resistance. XRD, SEM, wear test were used. Thermodynamic properties of this alloy were mentioned. The results of this work is quite interesting for the fundamental science and industry.

The manuscript is written clearly, well-structured and has a good scientific soundness.

I think, this manuscript can be published in the Entropy journal after minor revision taking into account and some of the remarks described below:

1.  Section 2. Materials and Methods: chemicals manufacturer and synthesis procedure should be specified.

2.  Which equipment was used for XRD measurements?

3.  It is not clear how thermodynamic data were used in the work? Were they only calculated or did they have experimental confirmations? In the part of the Results, thermodynamic data practically do not appear.

4.  It is not clear if studied material is real alloy, or heterogeneous mixture, or solid solution. It should be discussed more by compare of XRD and SEM data.

  1. Please explain how the vickers microhardness values were determined specifically for the various phases A, B, C? Were the phases taken under a microscope?
  2. The authors claim that “Table 3 also shows no significant differences in the composition of the phases of the three alloys in the AC condition” (line 402). However, it can be seen that the composition of phase B varies greatly for each alloy. For example, the copper content in phase B of H-0.5Nb and H-1Nb alloys is 4.59 and 5.44 at. %, respectively. For the H-0Nb alloy, the copper content is 10.21 at. %. There is a 2-fold difference in the metal content. A similar situation is observed for chromium and nickel. How can the authors explain this discrepancy.
  3. What is the error in determining the content of elements in each alloy? Why the data is given up to the second decimal place, and not up to the first.
  4.  The contents of the elements are given with varying degrees of rounding in Table 2. Please round the values of all elements equally.
  5. In figure 9, you can change the division value on the Y axis. This will make the figure more informative.

Author Response

Dear Reviewer,                                                                    August 20, 2022

I am very grateful for your comments and observations. Authors are agree with comments and observations, every comment and change has been made.
